# Maize OPR2 and LOX10 Mediate Defense against Fall Armyworm and Western Corn Rootworm by Tissue-Specific Regulation of Jasmonic Acid and Ketol Metabolism

**DOI:** 10.3390/genes14091732

**Published:** 2023-08-30

**Authors:** Pei-Cheng Huang, John M. Grunseich, Katherine M. Berg-Falloure, Jordan P. Tolley, Hisashi Koiwa, Julio S. Bernal, Michael V. Kolomiets

**Affiliations:** 1Department of Plant Pathology and Microbiology, Texas A&M University, College Station, TX 77843-2132, USA; pei-cheng.huang@ag.tamu.edu (P.-C.H.); katherine.berg@ag.tamu.edu (K.M.B.-F.); 2Department of Entomology, Texas A&M University, College Station, TX 77843-2475, USA; johngrunseich@tamu.edu; 3Department of Horticultural Sciences, Texas A&M University, College Station, TX77843-2133, USA; jordan.tolley@lonestar.edu (J.P.T.); hisashi.koiwa@ag.tamu.edu (H.K.)

**Keywords:** OPDA reductase, lipoxygenase, *Spodoptera frugiperda*, *Diabrotica virgifera virgifera*, ketols, death acids, jasmonic acid, maize

## Abstract

Foliage-feeding fall armyworm (FAW; *Spodoptera frugiperda*) and root-feeding western corn rootworm (WCR; *Diabrotica virgifera virgifera*) are maize (*Zea mays* L.) pests that cause significant yield losses. Jasmonic acid (JA) plays a pivotal defense role against insects. 12-oxo-phytodienoic acid (12-OPDA) is converted into JA by peroxisome-localized OPDA reductases (OPR). However, little is known about the physiological functions of cytoplasmic OPRs. Here, we show that disruption of *ZmOPR2* reduced wound-induced JA production and defense against FAW while accumulating more JA catabolites. Overexpression of *ZmOPR2* in *Arabidopsis* enhanced JA production and defense against beet armyworm (BAW; *Spodoptera exigua*). In addition, *lox10opr2* double mutants were more susceptible than either single mutant, suggesting that *ZmOPR2* and *ZmLOX10* uniquely and additively contributed to defense. In contrast to the defensive roles of *ZmOPR2* and *ZmLOX10* in leaves, single mutants did not display any alteration in root herbivory defense against WCR. Feeding on *lox10opr2* double mutants resulted in increased WCR mortality associated with greater herbivory-induced production of insecticidal death acids and ketols. Thus, *ZmOPR2* and *ZmLOX10* cooperatively inhibit the synthesis of these metabolites during herbivory by WCR. We conclude that *ZmOPR2* and *ZmLOX10* regulate JA-mediated resistance in leaves against FAW while suppressing insecticidal oxylipin synthesis in roots during WCR infestation.

## 1. Introduction

Maize (*Zea mays* L.) is a major cereal crop produced globally and a staple crop in many parts of the world. Maize is also a major feed source for livestock worldwide and is used to produce a variety of foods and industrial products. In 2021, U.S. maize production was worth close to USD 86 billion [1]. Maize production is constantly threatened by abiotic and biotic stressors, including the attack of insect herbivores [2]. Fall armyworm (FAW, *Spodoptera frugiperda*) and western corn rootworm (WCR, *Diabrotica virgifera virgifera*) are among the world’s most important maize pests [3,4,5,6,7].

FAW causes devastating yield losses annually in maize [8]. Its larvae feed on foliage, stems, and reproductive tissues, and maize is its preferred host. Aside from maize, it has been reported to attack >350 plant species, including numerous crops such as maize, sorghum, soybean, cotton, barley, and wheat [9,10]. FAW is native to the American tropics and subtropics and has recently invaded Africa, Asia, and Australia [8,10]. The rapid global spread of FAW has caused significant yield losses and threatened food security, especially in African countries where maize is a staple and the most susceptible crop. Annual yield losses caused by FAW are estimated to exceed USD 9 billion [11,12]. Resistance to FAW and other chewing insects is mediated mainly by jasmonic acid (JA)-mediated signaling [13,14,15].

WCR is a specialist herbivore of maize and the most important maize pest in the USA [16,17]. WCR larvae cause significant injury to maize roots and are difficult to control with insecticides because of their belowground feeding habits. WCR is proposed to have originated in Mexico or Guatemala and spread into North America with the spread of maize cultivation [18]. WCR was first identified as a pest in the USA in the early 1900s and quickly expanded its range throughout North America, where it has caused significant economic losses for over a century, especially in the USA corn belt [16,18]. WCR was detected in Europe in 1992 and has now invaded 21 European countries [19]. WCR is known as the billion-dollar beetle, because annual costs associated with its control and lost yield exceed USD 1 billion [20]. Maize resistance mechanisms against WCR are not well understood, and whether JA signaling plays any role in defense responses is unclear [21].

Lipoxygenases (LOXs) in plants incorporate molecular oxygen into linoleic acid (C18:2) and linolenic acid (C18:3) at either carbon position 9 or 13 of the 18-carbon chain and are functionally grouped into 9-LOXs and 13-LOXs [22,23]. Oxidized polyenoic fatty acids are collectively called oxylipins, and most of them are derivatives of seven downstream branches of lipoxygenase (LOX) pathways [22,23,24]. Jasmonates, including JA and its derivatives, are lipid-derived phytohormones that play pivotal roles in insect defense responses [25,26,27]. The initial step of JA biosynthesis occurs in chloroplasts, where 13-lipoxygenases (13-LOX) oxidize α-linolenic acid (C18:3) and involve the ensuing enzymatic activities of allene oxide synthase (AOS) and allene oxide cyclase (AOC) to produce 12-oxo-phytodienoic acid (12-OPDA). 12-OPDA is then converted to JA by OPDA reductase (OPR) in the peroxisomes [22,24,28]. The formation of the biologically active jasmonoyl-isoleucine (JA-Ile) is catalyzed by Jasmonate Resistant 1 (JAR1) in the cytoplasm [29]. In addition to JA, AOS produces ketols, a group of C18-oxylipin compounds with hormone-like signaling activities that were recently shown to play essential roles in defense against fungal pathogens and chewing insects [30,31,32,33,34,35]. Some ketols, including 9-hydroxy-10-oxo-12(*Z*), 15(*Z*)-octadecadienoic acid (9,10-KODA), 9-hydroxy-12-oxo-10(*E*), 15(*Z*)-octadecadienoic acid (9-12-KODA), and 9-hydroxy-12-oxo-10(*E*)-octadecenoic acid (9-12-KOMA), have been shown to prime plants for defense against pathogens [34,35]. Moreover, ketol 9 and 10-KODA displayed hormone-like and insecticidal activities in defense against FAW and suppressed FAW larvae growth [31].

Plant OPRs are phylogenetically and functionally classified into OPRI and OPRII subfamilies depending on their substrate catalytic activity [36,37]. OPRII subfamily peroxisome-localized enzymes are JA-producing OPRs because they preferentially catalyze the conversion of the natural JA precursor, *cis*-(+)-OPDA. The members of the cytosolic OPRI subfamily were found to reduce *cis*-(-)-OPDA and were long believed to be irrelevant in JA biosynthesis [38,39]. So far, several OPRII genes have been characterized. Arabidopsis intronic T-DNA insertional *opr3* mutants are deficient in wound-induced JA production and more susceptible to cabbage looper caterpillars [40]. Disruption of both maize *ZmOPR7* and *ZmOPR8* results in JA deficiency and extreme susceptibility to the root-rotting oomycete *Pythium* spp. and beet armyworm (BAW, *Spodoptera exigua*) [13]. Little is known about the biochemical and physiological functions of OPRI subfamily enzymes, and it is unknown whether they contribute to insect defense.

Previously, the OPRI subfamily member ZmOPR2 was shown to be localized to the cytoplasm [41] and functions in salicylic acid (SA)-mediated defense responses against maize biotrophic and hemibiotrophic pathogens via suppression of JA production in response to pathogen infection [42]. However, its role in wound-induced JA production and insect defense remains unexplored. This study aimed to partly fill this gap and explore the role, if any, of ZmOPR2 in insect defense in both above- and belowground tissues. We show here that *ZmOPR2* contributes to aboveground defense against FAW by regulating wound-induced JA production and catabolism and interacting additively with *ZmLOX10* in resistance to herbivory in leaves. Additionally, we show that while single *opr2* or *lox10* mutants did not affect maize defense against WCR, *lox10opr2* double mutants markedly reduced the survival of WCR larvae. Metabolite profiling of the *lox10opr2* double mutants under WCR herbivory revealed increased accumulations of multiple ketols and death acids known to have insecticidal activities when delivered via an artificial diet. Given these findings, we concluded that *ZmOPR2* and *ZmLOX10* together negatively regulate the production of insecticidal death acids and ketols, which are likely a part of the host defense strategy against herbivory by WCR. Additionally, our data suggested that JA may not have as significant a function in herbivory defense in maize roots as it has in the leaves.

## 2. Materials and Methods

### 2.1. Plant Materials

Mutator-insertional *opr2-1* (PV 03 80 A-05) and *opr2-3* (mu1079063:Mu; stock ID UFMu-08953) alleles were backcrossed to the B73 inbred line and advanced to BC_7_ and BC_1_ stages, respectively. Mutants and their corresponding near-isogenic wild-types were identified from F2 segregating populations by PCR genotyping using *Mu*-terminal inverted repeat-specific and gene-specific primers as described previously [42]. *opr2-1* and *opr2-3* are knockout alleles and were described in detail in Huang et al. [42]. The identification of the *lox10-3* allele was previously described by Christensen et al. [14]. The double mutant *lox10opr2* was generated by crossing the single mutants *lox10-3* and *opr2-1* at the BC_7_ genetic stage [42]. Maize seeds were grown in conical pots (20.5 by 4 cm) filled with commercial potting mix (Jolly Gardener Pro Line C/20 potting mix) on light shelves at room temperature (22 to 24 °C) with a 16 h light period. For generating Arabidopsis *ZmOPR2*-overexpression lines, gateway vectors carrying the *ZmOPR2* gene were transformed into the *rdr6-11* background as described in Tolley et al. [41] and grown under the same conditions. Two T4 homozygous ZmOPR2-GFP overexpression lines, *#32-9* and *#39-1*, were named in this study *OE1* and *OE2* lines, respectively, and untransformed *rdr6-11* plants were used as the WT controls.

### 2.2. Fall Armyworm Bioassays

The laboratory strain of FAW (*Spodoptera frugiperda*) was purchased from Benzon Research (Carlisle, PA, USA). The eggs laid on paper towels were hatched and reared on an artificial diet purchased from Southland Products Inc. (Southland Products Inc, Lake Village, AR, USA). To evaluate insect resistance and leaf damage area, the fourth leaf of maize plants at the V4 stage were individually caged and infested with one 3rd instar FAW larva per spot for approximately one hour, then moved toward the base afterward, and so on. Leaves were scanned, and eaten areas were measured with ImageJ software (https://imagej.nih.gov/ij/index.html, accessed on 8 August 2023). Larval weight gain experiments were performed by caging six FAW neonates with individual maize plants at the V3 developmental stage and allowing them to move and feed on the plant freely for 7 days. FAW larvae were removed from the plants, and total weight was determined 7 days post-infestation.

### 2.3. Beet Armyworm Bioassay

BAW (*Spodoptera exigua*) eggs were purchased from Benzon Research (Carlisle, PA, USA). After hatching, neonate larvae were reared on an artificial diet purchased from Southland Product Inc. (Lake Village, AR, USA) for 5 days. Third instar larvae were starved overnight, and the initial weight was measured before transferring to a 0.8% agar plate. Mature rosette leaves were cut from 4-week-old *Arabidopsis* plants and placed on the plate with 4 BAW larvae per plate. Fresh leaves were provided two days after the initial feeding, and the experiment was terminated on day 4. The larval weight gain was determined after subtracting the initial weight and normalizing it by the number of recovered larvae.

### 2.4. WCR Bioassays

WCR (*Diabrotica virgifera virgifera*) eggs were provided by the USDA-ARS North Central Agriculture Research Laboratory (Brookings, SD, USA) and stored at 4 °C until use. The diapausing Tent strain was used for this study. Prior to use, eggs were washed and incubated on moist filter paper for 12 days at 27 °C. Neonates were used within 24 h after the first emergence. For the WCR assay, maize seeds were grown in conical pots (20.5 by 4 cm) filled with commercial potting mix (Jolly Gardener Pro Line C/25 potting mix) in a climate-controlled, insect-free growth room under artificial full-spectrum growing lights (27 °C, 50% relative humidity, 12:12 h light:dark cycle, PPFD 450 µmol m^−2^ s^−1^). Experiments were performed in the same environmental conditions in which plants were grown. Plants were watered as needed before the application of WCR neonates. Two to three week old maize plants that had 3 fully expanded true leaves were infested with 10 WCR neonates. WCR larvae were removed from the soil, and total weight gain was determined 10 days post-infestation. The recovery of WCR larvae is a proxy measure of survivorship. WCR head capsules were then imaged and measured using ImageJ to quantify larval instar stage (ImageJ, National Institute of Health, Bethesda, MD, USA). The soil was then removed from the root tissue, and the fresh root weight was measured. Root ratio was analyzed as larvae damaged over undamaged root mass to determine host plant tolerance as compared to WT.

### 2.5. Oxylipin Profiling of Wounded Leaf Tissue and WCR-Infested Root Tissues

For wounding treatment in leaves, the third fully expanded leaves of seedlings at the V3 developmental stage were wounded seven times using a hemostat, with three wound sites on one side and four on the other side of the midvein and wound sites approximately 1 cm apart in the middle portion of the leaf. For Arabidopsis, rosette leaves were wounded twice using forceps across the midvein. The wounded regions were then harvested in 2 mL screwcap Fast-Prep tubes (Qbiogene, Carlsbad, CA, USA) in liquid nitrogen and stored in an −80 °C freezer. For WCR-infested root tissues, maize root tissues were washed at 0 (CTRL), 8, 24, and 48 h after being exposed to WCR neonates, harvested in liquid nitrogen, and stored in an −80 °C freezer. Phytohormone and oxylipin extraction and profiling of wounded leaf tissue were performed using a LC-MS/MS as described in Huang et al. [42].

### 2.6. Statistical Methods

Statistical analyses were performed using the software programs R (R version 4.2.2, R Core Team, 2023, Vienna, Austria), JMP Pro 17 (SAS Institute Inc., Cary, NC, USA), and GraphPad Prism 10 (GraphPad Software, Boston, MA, USA). FAW leaf area consumed and larval weight gain were analyzed using the Student’s *t*-test. BAW larval weight gain was analyzed using ANOVA followed by Dunnett’s test. Two-way ANOVAs, with genotype and time as the two factors, were used for leaf metabolite analysis of WT-mutant comparisons followed by Sidak’s multiple comparisons test for comparisons within time. FAW larval weight gain and leaf area consumed were compared using ANOVA followed by Tukey’s honestly significant differences (HSD) test. Root and shoot ratios, WCR larvae recovery, and WCR mass gain were analyzed using ANOVA followed by Tukey’s multiple comparison test. Larval instars were compared using ANOVA, followed by Dunnett’s multiple comparison test. Root hormone analyses at 24 hpi were compared using the Student’s *t*-test.

## 3. Results

### 3.1. ZmOPR2 Promotes Defense against FAW Herbivory

In a previous study, we reported that *ZmOPR2* functions in SA-mediated defense responses against (hemi)biotrophic pathogens by suppressing the activity of JA-producing ZmLOX10 [42]. In this study, we tested whether *ZmOPR*2 is relevant to insect defense by caging single 3rd instar FAW larvae on the leaves of *opr2-1* and *opr2-3* mutants and their respective WTs at the V4 developmental stage in maize. After 6 h, FAW larvae consumed significantly more leaf tissue in *opr2-1* and *opr2-3* mutant seedlings (~1.5- and 3.5-fold larger, respectively) compared to consumption in WT seedlings, suggesting that *opr2* mutants are more susceptible to FAW (Figure 1A). Consistent with this result, FAW larvae weighed significantly more (>40%) after feeding on *opr2-1* and *opr2-3* mutants for 7 days than those fed on WT plants (Figure 1B). These results suggested that *ZmOPR2* is involved in defense against aboveground herbivory by chewing insects. Transgenic Arabidopsis plants overexpressing *ZmOPR2* were generated to corroborate whether *ZmOPR2* enhances insect defense [41]. Third instar BAW larvae were offered mature rosette leaves from *ZmOPR2*-OE Arabidopsis lines or control plants for 4 days. After 4 days, BAW larvae gained significantly less weight (29–39% reduction) after feeding on *ZmOPR2*-OE line leaves compared to larvae feeding on WT leaves, suggesting that overexpression of *ZmOPR2* in Arabidopsis enhanced defense against BAW (Figure 1C). Together, these results showed that *ZmOPR2* promotes defense in aboveground tissues against chewing insects such as FAW and BAW.

### 3.2. ZmOPR2 Enhances Wound-Induced JA Accumulation

A previous study showed that disruption of *ZmOPR2* in maize resulted in reduced resistance to hemibiotrophic *Colletotrichum graminicola* and was associated with greater infection-induced JA accumulation [42]. Surprisingly, in the present study, we found that *opr2* mutants were more susceptible to FAW, which prompted us to test whether *ZmOPR2* plays a role in wound-induced JA production. To this end, leaves of *opr2-1* and *opr2-3* mutants and WT were mechanically wounded, and the accumulation of jasmonates was measured at 0, 1, and 2 h post-wounding (hpw). We found that *opr2-1* and *opr2-3* mutants accumulated significantly lower levels of wound-induced JA at 1 and 2 hpw (16 and 27% reduction in *opr2-1*, 31 and 34% reduction in *opr2-3*, respectively) (Figure 2B) and JA-Ile (25 and 41% reduction in *opr2-1*, 42 and 55% reduction in *opr2-3*) (Figure 2C). Both alleles accumulated similar amounts of 12-ODPA compared to WT (Figure 2A).

In support of the notion that *ZmOPR2* promotes wound-induced JA synthesis, we found that Arabidopsis *ZmOPR2*-OE lines accumulated significantly higher levels of JA at 0.5 and 1 hpw (42% and 49% increase, respectively) (Figure 3C) and JA-Ile (~20% and >70% increase) (Figure 3D) while accumulating less dn-12-OPDA (~40% and >20% reduction) (Figure 3B) compared to WT. We found no significant difference in the levels of 12-OPDA (Figure 3A).

Accumulating evidence suggests that JA turnover and homeostasis are tightly regulated by JA catabolism in a stress-specific manner [28]. Therefore, we also measured the levels of JA catabolites, including 12-hydroxy-jasmonic acid (12OH-JA), 12-hydroxy-jasmonyl-L-isoleucine (12OH-JA-Ile), and 12-carboxy-jasmonoyl-L-isoleucine (12COOH-JA-Ile), in leaves upon wounding. Interestingly, we found that *opr2-1* and *opr2-3* mutants accumulated significantly higher levels of JA catabolites at 1 and 2 hpw, including 12-OH-JA (73 and 51% increase in *opr2-1*, 6 and 69% increase in *opr2-3*, respectively), 12-OH-JA-Ile (40 and 18% increase in *opr2-1*, 96 and 90% increase in *opr2-3*), and 12-COOH-JA-Ile (28% reduction and 107% increase in *opr2-1*, 280 and 78% increase in *opr2-3*) compared to WT (Figure 4A–C). These results suggest that *ZmOPR2* not only regulates the biosynthesis of biologically active JA but also JA catabolism after wounding. To corroborate these results, we also measured levels of JA catabolites in mechanically wounded Arabidopsis *ZmOPR2*-OE lines at 0.5 and 1 hpw. We found that at 0.5 and 1 hpw after wounding, significantly lower levels of 12-OH-JA (30–37% and 22–46% reduction, respectively), 12-OH-JA-Ile (38–45% and 17–22% reduction), and 12-COOH-JA-Ile (51–53% reduction at 1 hpw) were accumulated in *ZmOPR2*-OE lines compared to WT (Figure 4D–F). Together, these results showed that *ZmOPR2* enhances wound-induced JA and JA-Ile biosynthesis and regulates the catabolism of jasmonates, the first report of such a specific effect of any plant OPR on JA catabolism.

### 3.3. ZmOPR2 and ZmLOX10 Have an Additive Effect on Defense against FAW

Previous reports showed that *ZmLOX10* plays an important role in plant modulation of insect defense by taking part in the biosynthesis of green leaf volatiles (GLV) and JA in response to wounding and insect herbivory [14,43]. Although mutation of *ZmOPR2* increased the production of ZmLOX10-mediated GLVs and accumulation of JA in response to pathogen infection [42], similar to *lox10* mutants, *opr2* mutants in the present study accumulated less wound-induced JA and JA-Ile and were more susceptible to a chewing insect, FAW. To explore the relative contributions of *ZmOPR2* and *ZmLOX10* to defense against herbivory, we performed FAW bioassays on single *opr2-1* and *lox10-3* mutants, *lox10opr2* double mutants, and corresponding WT controls. We found that while the 3rd instar FAW larvae consumed 64% and 60% more leaf tissue in single *opr2-1* and *lox10-3* mutants, respectively, compared to WT, FAW larvae consumed twice as much (126%) *lox10opr2* double mutant tissue (Figure 5A). Correspondingly, FAW larvae gained more weight after feeding on single *opr2-1* and *lox10-3* mutants (48% increase) compared to those fed on WT, and FAW larvae that fed on *lox10opr2* double mutants showed the greatest increase in weight (167%) relative to WT (Figure 5B). Together, these results suggested that *ZmOPR2* and *ZmLOX10* acted additively in defense modulation against the chewing insect FAW and that the contribution of *ZmOPR2* to defense against chewing insects extends beyond its role in JA synthesis.

### 3.4. ZmOPR2 and ZmLOX10 Negatively Regulate Production of Insecticidal Oxylipins during Root Herbivory by WCR

Multiple lines of evidence, including the current study, confirm that JA plays an essential role in regulating plant defense responses against chewing insects in aboveground tissues [13,44]. However, whether JA is required for WCR defense in roots remains largely unknown. Here, we tested whether *ZmOPR2* and *ZmLOX10* contribute to JA synthesis and defense against chewing insects in roots by assaying corresponding single and double mutants. We found that the survivorship of WCR larvae feeding on the roots of *lox10opr2* mutants was lower in comparison to the survivorship of larvae feeding on the roots of single *opr2-1* or WT plants (ANOVA, F3, 119 = 6.11, *p* < 0.001) (Figure 6A), but showed no difference in comparison to a single *lox10-3* mutant. However, no significant difference in larval weights was found after larvae fed on *opr2-1*, *lox10-3*, or *lox10opr2* mutants compared to larvae fed on WT (ANOVA, F3,95 = 0.427, *p* = 0.733) (Figure 6B). Additionally, WCR larvae feeding on *lox10-3* or *lox10opr2* mutants developed slower compared to those feeding on *opr2-1* mutant and WT plants (ANOVA, F3, 45 = 4.428, *p* = 0.008) (Figure 6C). Finally, *lox10opr2* mutants showed greater tolerance (ability to recover tissues lost to herbivory) compared to WT, *opr2-1*, and *lox10-3* mutants (ANOVA, F3, 119 = 3.907, *p* = 0.011), as indicated by their greater root mass ratio (ratio of root mass of plants exposed to WCR relative to root mass of plants not exposed to WCR) (Figure 6D). Together, these results showed that *ZmOPR2* and *ZmLOX10* downregulated both direct defense and tolerance against the chewing insect WCR in maize roots.

To investigate metabolites responsible for the enhanced defense and greater tolerance of *lox10opr2*, we quantified root metabolites of WT and *lox10opr2* double mutant plants at 0 (no WCR herbivory), 8, 24, and 48 h post-exposure to WCR larvae. The relative abundances of root metabolites in WT and *lox10opr2* double mutant plants were visualized in a two-way dendrogram and heatmap to reveal the temporal dynamics and variation of metabolite accumulation (Appendix A). Examination of the heatmap suggested that, in aggregate, metabolite levels in WT decreased between 0 and 8 h and remained at low levels through 48 h after exposure to WCR larvae, while aggregate metabolite levels in *lox10opr2* double mutants were low between 0 and 8 h, increased at 24 h, and decreased at 48 h (Appendix A). Interestingly, accumulation of JA and JA-Ile did not differ between WT and *lox10opr2* double mutants at any of the time points (0–48 h), suggesting that, while phenotypic changes were evident in the double mutants, these changes were not due to changes in JA-related abundances (Appendix A). After paired comparisons between WT-*lox10opr2* mutants for all metabolites at 24 h post exposure to WCR, we identified several compounds that accumulated at significantly higher levels in *lox10opr2* mutant plants than WT at 24 h post WCR infestation (Figure 7A). These included four oxylipins synthesized by the 9-LOX pathways: 10-OPDA (Student’s *t*-test, T = 2.583, *p* = 0.036), 10-OPEA (Student’s *t*-test, T = 2.899, *p* = 0.023), 13,10-KODA (Student’s *t*-test, T = 3.667, *p* = 0.014), and 13,10-KOMA (Student’s *t*-test, T = 2.793, *p* = 0.038) and two oxylipins synthesized in the 13-LOX pathways: 13,12-KODA (Student’s *t*-test, T = 2.546, *p* = 0.034) and 12,13-EpOD (Student’s *t*-test, T = 2.497, *p* = 0.037) (Figure 7B); and ABA (Student’s *t*-test, T = −3.0192, *p* = 0.016) (Appendix A) relative to their WT controls before returning to WT levels at 48 h post-infestation. It is likely that these insecticidal and signaling oxylipins [31,45] underlie the increased mortality of larvae feeding on *lox10opr2* roots and enhanced tolerance of *lox10opr2* double mutants.

## 4. Discussion

JA has been extensively studied in aboveground tissue for its role in herbivory defense against chewing insects in numerous plant species, including maize [13,14,46,47]. For instance, suppression of JA-producing OPRII enzymes resulted in reduced resistance to insect herbivory in aboveground *Arabidopsis* and maize tissues [13,40]. In contrast, there are no studies reporting the roles of any OPRI subfamily member in insect defense. The maize genome contains eight OPR genes, six of which belong to the OPRI subfamily [37]. Green leaf volatiles, mechanical wounding, and insect elicitor treatment have been shown to induce expression of *ZmOPR1/2* [37,48], and feeding by either BAW or FAW induced *ZmOPR2* expression in maize [15,49], suggesting that *ZmOPR2* may be involved in insect defense. Elevated pathogen-induced JA contents in *opr2* mutants [42] led us to test whether *opr2* mutants also produce higher wound-induced jasmonates and are more resistant to FAW. However, the results showed that *opr2* mutants unexpectedly accumulated lower levels of wound-induced JA and JA-Ile (Figure 2B,C), which was associated with increased susceptibility to FAW as manifested by greater consumption of leaf tissue by FAW and greater weight gain in FAW larvae after feeding on *opr2* mutants compared to those feeding on WT. Correspondingly, overexpression of *ZmOPR2* in *Arabidopsis* enhanced the production of wound-induced JA and JA-Ile (Figure 3C,D) and resistance to BAW. ZmOPR2 belongs to the OPRI subfamily that preferentially catalyzes the reduction of *cis*-(-) OPDA over the JA precursor *cis*-(+) OPDA [36] and localizes in the cytoplasm [41]. Therefore, ZmOPR2 was believed to not be directly involved in providing substrate for JA biosynthesis. In maize, only ZmOPR7 and ZmOPR8 belong to the OPRII subfamily, which preferentially catalyzes *cis*-(+) OPDA over *cis*-(-) OPDA to form the JA precursor OPC 8:0, and disruption of these two genes results in JA-deficiency and increased susceptibility to chewing insects and necrotrophic pathogens [13,50]. However, there are trace amounts of JA accumulation in the young leaves of mature *opr7opr8* plants [13], suggesting the occurrence of at least another OPR enzyme capable of producing JA under specific conditions. In *Arabidopsis*, the OPRII enzyme AtOPR3 was considered the only JA-producing OPR enzyme [51,52]. AtOPR2, an OPRI subfamily member, was ruled out of playing a direct role in JA biosynthesis because it is not localized to the peroxisome and because of its low efficiency in catalyzing reduction on *cis*-(+) OPDA. Recently, a novel AtOPR3-independent pathway for JA biosynthesis was discovered that involves cytosolic AtOPR2 and uses 4,5-didehydrojasmonate as a JA substrate derived from dn-12-OPDA rather than *cis*-(+) OPDA [53]. In addition, increased dosage or transgenic overexpression of OPRI subfamily genes reduced seminal root growth in wheat and was associated with a higher accumulation of JA [54]. In agreement with these findings, we observed that *ZmOPR2*-OE lines accumulated reduced levels of wound-induced dn-12-OPDA, presumably due to its utilization as the primary substrate for JA synthesis (Figure 3B). Thus, it is likely that ZmOPR2 is also involved in wound-induced JA production in the pathway identified for *Arabidopsis* AtOPR2. In maize, the existence of a similarly alternative JA synthesis pathway is supported by our previous finding that normally JA-deficient *opr7opr8* mutants accumulated significant levels of JA in response to *C. graminicola* at 7 days post-infection [55].

Other than directly producing JA, the decreased wound-induced JA found in *opr2* mutants may be due to increased catabolism of JA based on our finding that *opr2* mutants accumulated significantly higher levels of JA catabolites, such as 12-OH-JA, 12-OH-JA-Ile, and 12-COOH-JA-Ile (Figure 4A–C). In contrast, after *C. graminicola* infection, susceptible *opr2* mutants showed greater levels of JA and JA-Ile [42], accompanied by significantly lower pathogen-induced accumulation of 12-OH-JA (Appendix A), while 12-OH-JA-Ile and 12-COOH-JA-Ile were undetectable. Together, these data suggest that *ZmOPR2* may not only contribute to JA biosynthesis directly but also regulate JA catabolism under various stress conditions by a yet unknown mechanism.

Our results provided evidence that, in addition to its important role in JA synthesis and catabolism, *ZmOPR2* appears to contribute to defense by an additional yet not understood mechanism. This conclusion is based on our finding that *lox10opr2* double mutants are more susceptible than single *lox10* or *opr2* mutants (Figure 5). This result suggests that *ZmOPR2* and *ZmLOX10* interact positively to regulate defense against chewing insects, and the increased susceptibility of the double mutant cannot be explained by lower JA or GLV production alone, as reported for *lox10-3* single mutants [14]. Based on the well-reported biochemical activities of the OPRI subfamily, ZmOPR2 may also contribute to defense by detoxifying multiple α, β-unsaturated carbonyls [56].

Plants are likely to orchestrate divergent defense responses to aboveground and belowground insect herbivory. Contrasting results have been shown in defense against FAW and WCR in Mp708, a FAW-resistant inbred line that was found to be susceptible to WCR [57,58]. While JA and related metabolites play essential roles in the mediation of defense responses against herbivory in aboveground tissues, several studies reported a much lower degree of herbivory- or wounding-induced accumulation of JA and JA-related metabolites and lower expression of JA-dependent genes in infested roots [59,60,61]. *ZmOPR1* and *ZmOPR2* are both significantly induced after WCR infestation, while expression of JA-producing *ZmOPR7* and *ZmOPR8* is not affected or minimally so per RNAseq transcriptome analyses of maize-WCR interactions [59,61]. Additionally, in the present study, we showed that JA accumulation upon WCR infestation was not significantly changed through 48 h post-infestation, suggesting that *ZmOPR1* and *ZmOPR2* may contribute to resistance to WCR via a JA-independent pathway and that JA may have little relevance to WCR defense. Given the clear evidence presented here that both *ZmOPR2* and *ZmLOX10* are required for defense against leaf herbivory by FAW, it was surprising to find that neither *lox10-3 nor opr2-1* single mutants were affected in their resistance to WCR, but *lox10opr2* double mutants were more tolerant to herbivory, as evidenced by less tissue consumed by WCR larvae. Notably, feeding on *lox10opr2* double mutant roots resulted in greater mortality of WCR larvae. In the metabolite analysis of roots in response to WCR infestation, we found that the more tolerant *lox10opr2* mutants accumulated higher levels of several oxylipins at 24 h post-WCR infestation, including death acids, 10-OPDA and 10-OPEA, and several ketols, 13,10-KODA, 13,10-KOMA, and 13,12-KODA (Figure 7). A greater accumulation of these metabolites may underlie the greater mortality of WCR larvae that we found. 10-OPDA and 10-OPEA possess insecticidal activity and strongly suppress larval growth [45]. For instance, fatty acid-derived ketols were shown to contain hormone-like signaling activities to induce systemic resistance against pathogens and insects [30,31,34,35]. Moreover, Yuan et al. [31] showed that in addition to strong signaling activity, the ketol 9,10-KODA displayed insecticidal activity against FAW. Whether the ketols 13,10-KODA, 13,10-KOMA, and 13,12-KODA that accumulated to greater levels in *lox10opr2* mutants also possess insecticidal activities or suppress larval growth requires further study. Nevertheless, these data suggest that *ZmOPR2* and *ZmLOX10* negatively regulate WCR resistance by suppressing the biosynthesis of lipid-derived insecticidal death acids and ketols.

## 5. Conclusions

In this study, we showed that disruption of *ZmOPR2* resulted in reduced accumulation of wound-induced JA, elevated levels of JA derivatives, and increased susceptibility to leaf herbivory by FAW. Overexpression of *ZmOPR2* in *Arabidopsis* enhanced wound-induced JA production, suppressed JA catabolism, and increased resistance to BAW. Together, these data demonstrated that ZmOPR2, a member of the OPRI subfamily enzymes, modulates defense against leaf herbivores by regulating JA homeostasis. Moreover, we found that *ZmOPR2* and *ZmLOX10* synergistically contribute to insect defense in the aboveground tissue. In contrast, *ZmOPR2* and *ZmLOX10* cooperatively suppress resistance to WCR by inhibiting the synthesis of death acids and ketols in roots upon WCR infestation. The results suggest that these metabolites may play a major role in the defense against root herbivory.

## Figures and Tables

**Figure 1 genes-14-01732-f001:**
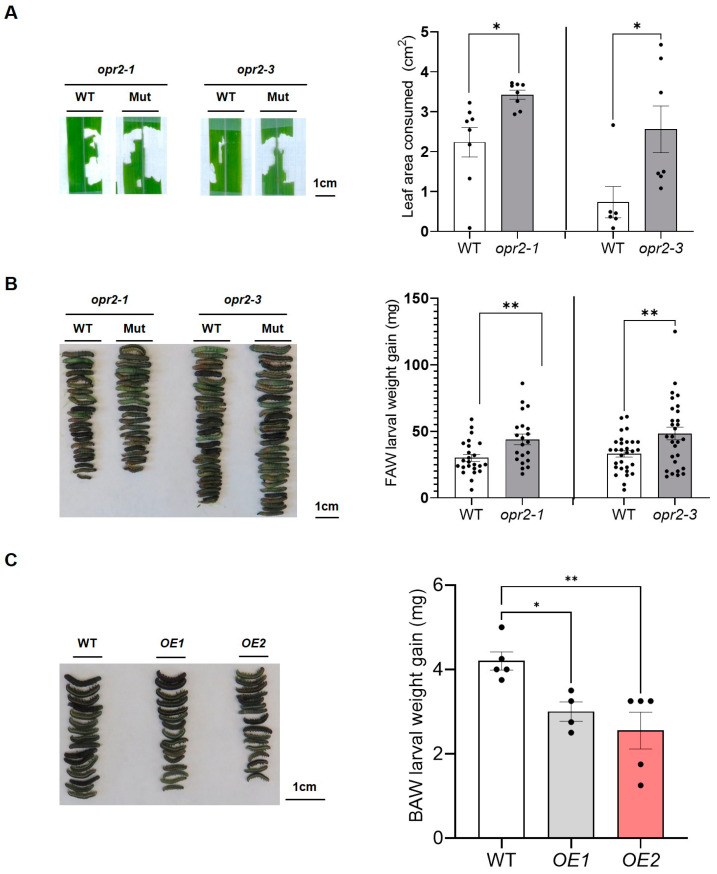
*ZmOPR2* promotes insect resistance. (**A**) The fourth leaf of *opr2-1* and its WT at V4 stage were caged and infested with one 3rd instar FAW larva per leaf. The leaves were scanned after the experiment, and eaten leaf areas were measured using ImageJ. (**B**) *opr2-*1 and *opr2-3* mutants and their respective WT plants at V3 stage were caged with 6 FAW neonates per plant. FAW larvae were removed from the plants, and larval weight was analyzed 7 days post-infestation. Bars means ± SE (*n* ≥ 25 FAW larvae removed from 5 different maize plants). Asterisks represent statistically significant differences between WT and mutant (Student’s *t*-test, * *p* < 0.05, ** *p* < 0.01). (**C**) BAW larval weight gain after feeding on *ZmOPR2*-OE lines and WT for 4 days. Gray and red bars represent two independent *ZmOPR2* overexpression lines. Bars are means ± SE (*n* ≥ 4 replicates of BAW larvae feeding on leaves from different Arabidopsis plants). Asterisks represent statistically significant differences (Dunnett’s test, * *p* < 0.05, ** *p* < 0.01).

**Figure 2 genes-14-01732-f002:**
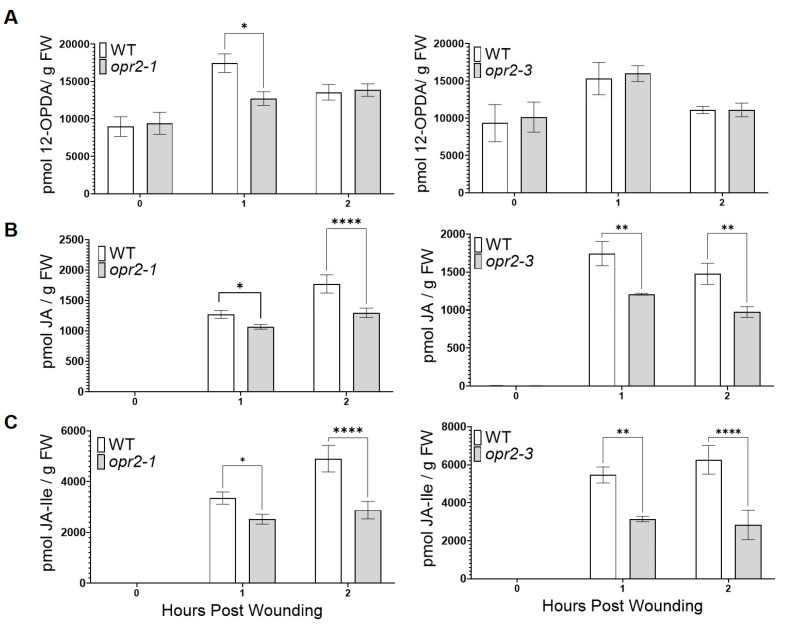
*opr2* mutants accumulated lower levels of wound-induced JA and JA-Ile. Contents of (**A**) 12-OPDA, (**B**) JA, and (**C**) JA-Ile were measured at 0, 1, and 2 h post-wounding (*n* ≥ 4 maize plants per genotype). Asterisks represent statistically significant differences between WT and mutant (Sidak’s multiple comparisons test, * *p* < 0.05, ** *p* < 0.01, **** *p* < 0.0001).

**Figure 3 genes-14-01732-f003:**
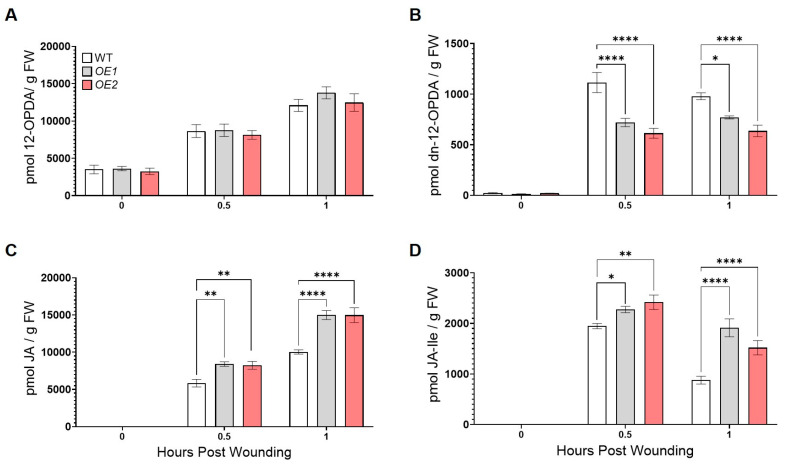
*ZmOPR2*-OE lines accumulated higher levels of wound-induced JA and JA-Ile. Contents of (**A**) 12-OPDA, (**B**) dn-12-OPDA, (**C**) JA, and (**D**) JA-Ile were measured at 0, 0.5, and 1 h post-wounding. Bars are means ± SE (*n*≥ 5 plants per genotype). Asterisks represent statistically significant differences (Sidak’s multiple comparisons test, * *p* < 0.05, ** *p* < 0.01, **** *p* < 0.01).

**Figure 4 genes-14-01732-f004:**
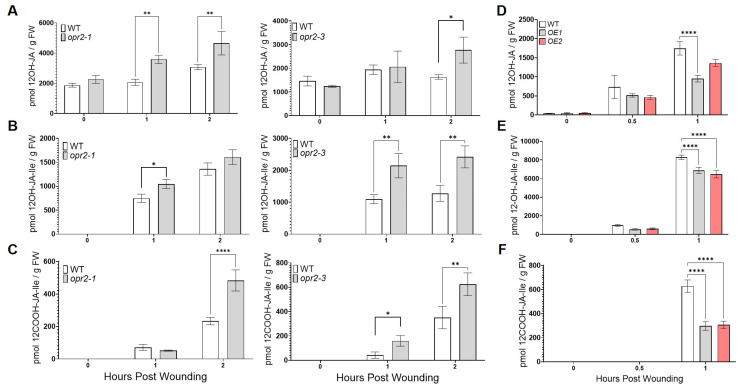
*ZmOPR2* regulates accumulation of JA catabolites in response to wounding. Contents of (**A**) 12OH-JA, (**B**) 12OH-JA-Ile, and (**C**) 12COOH-JA-Ile in *opr2-1* and *opr2-3* mutants and WT were measured at 0, 1, and 2 h post-wounding. Accumulation of (**D**) 12OH-JA, (**E**) 12OH-JA-Ile, and (**F**) 12COOH-JA-Ile was measured in ZmOPR2-OE lines and control plants at 0, 0.5, and 1 h post-wounding. Bars are means ± SE (*n* ≥ 4 plants per genotype). Asterisks represent statistically significant differences (Sidak’s multiple comparisons test, * *p* < 0.05, ** *p* < 0.01, **** *p* < 0.0001).

**Figure 5 genes-14-01732-f005:**
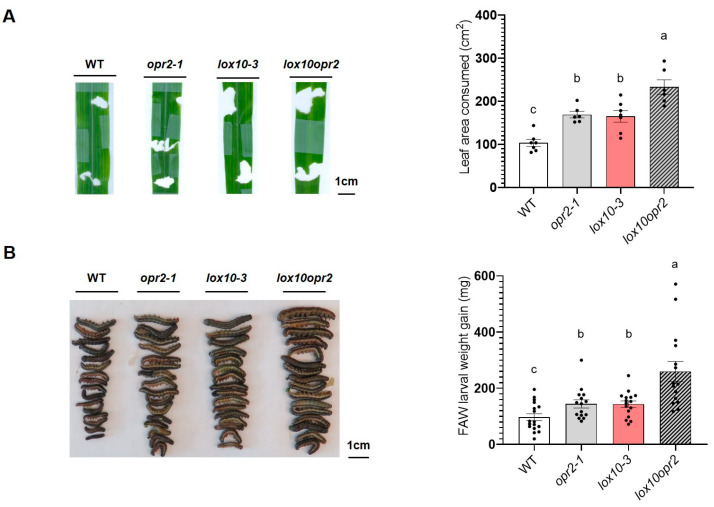
*ZmOPR2* and *ZmLOX10* act synergistically in the regulation of defense against FAW. (**A**) Average leaf consumed area of *opr2-1* (gray bar), *lox10-3* (red bar), and *lox10opr2* (slashed bar) mutants and their WT after FAW infestation. Eaten leaf area is the sum of all the damaged areas on one leaf. Bars are means ± SE (*n* ≥ 6 maize plants). (**B**) Average FAW larval weight gain after feeding on opr2-1 (gray bar), *lox10-3* (red bar), and *lox10opr2* (slashed bar) mutants and WT for 7 days. Bars are means ± SE (*n* ≥ 20 larvae from 5 different plants). Different letters indicate statistically significant differences in log-transformed data (Tukey’s HSD test, *p* < 0.05).

**Figure 6 genes-14-01732-f006:**
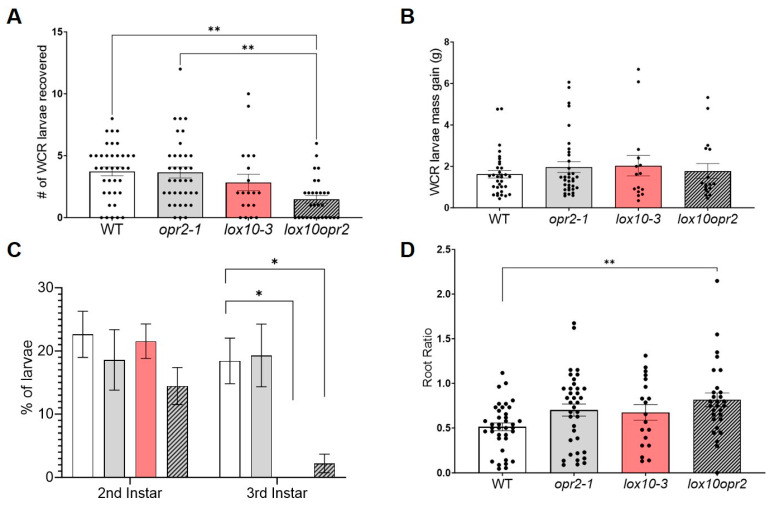
Mutations of both *ZmOPR2* and *ZmLOX10* suppressed WCR larval survival and development. (**A**) Larvae recovery and (**B**) Larval mass were recorded 10 days post-infestation, and (**C**) the developmental stages of the larvae recovered were determined by measuring larvae head capsules using ImageJ (**D**) Root ratio was analyzed as larvae damaged over undamaged root mass to determine tolerance in *opr2-1* (gray bar), *lox10-3* (red bar), and *lox10opr2* (slashed bar) mutants after comparing to WT. Bars are means ± SE. Asterisks represent statistically significant differences (**A**,**B**,**D**—Tukey HSD and **C**—Dunnett’s Test, * *p* < 0.05, ** *p* < 0.01).

**Figure 7 genes-14-01732-f007:**
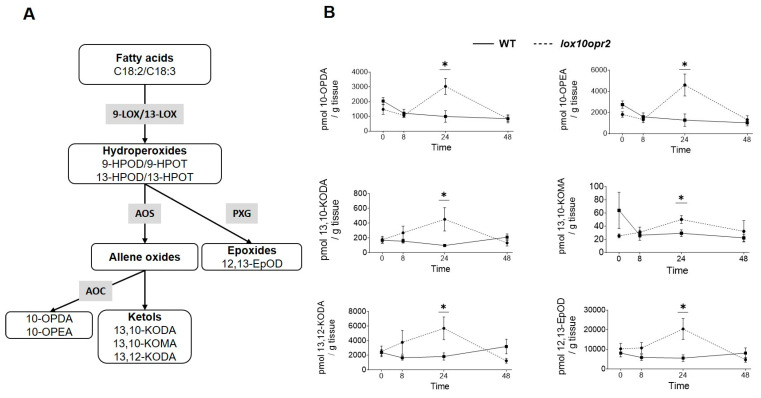
Disruption of both *ZmOPR2* and *ZmLOX10* enhances ketol and insecticidal compound production upon WCR infestation (**A**) Simplified lipoxygenase biosynthesis pathways showing the oxylipins with differential accumulation in the WT and *lox10opr2* double mutants. Hydroperoxides derived from C18:2/C18:3 by 9- or 13-LOXs are used as substrates for the downstream branches. Enzyme symbols are represented by gray boxes, while oxylipins are in white boxes and grouped by chemical class. Abbreviations are as follows: (Enzymes) AOC, allene oxide cyclase; AOS, allene oxide synthase; LOX, lipoxygenase; PXG, peroxygenase. (Oxylipins) C18:2, linoleic acid; C18:3, linolenic acid; 9-HPOD, (9*S*,10*E*,12*Z*)-9-hydroperoxy-10,12-octadecadienoic acid; 9-HPOT, (9*S*,10*E*,12*Z*,15*Z*)-9-hydroperoxy-10,12,15-octadecatrienoic acid; 10-OPDA, 10-oxo-11(*Z*),15(*Z*)-phytodienoic acid; 10-OPEA, 10-oxo-11(*Z*)-phytoenoic acid; 12,13-EpOD, cis-12,13-expoxy-9(*Z*),15(*Z*)-octadecenoic acid; 13,10-KODA, 13-hydroxy-10-oxo-11(*E*),15(*Z*)-octadecadienoic acid; 13,10-KOMA, 13-hydroxy-10-oxo-11(*E*)-octadecenoic acid; 13,12-KODA, 13-hydroxy-12-oxo-9(*Z*),15(*Z*)-octadecadienoic acid. (**B**) Accumulation of various oxylipins, including 10-OPDA, 10-OPEA, 13,10-KODA, 13,10-KOMA, 13,12-KODA, and 12,13-EpOD in WT and *lox10opr2* mutans at 0, 8, 24, and 48 h post-WCR infestation. Bars are means ± SE. Asterisks represent statistically significant differences (Student’s *t*-test, * *p* < 0.05).

## Data Availability

The datasets used and analyzed during the current study are available from the corresponding author upon reasonable request.

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
