# Peer review of "Maize OPR2 and LOX10 Mediate Defense against Fall Armyworm and Western Corn Rootworm by Tissue-Specific Regulation of Jasmonic Acid and Ketol Metabolism"

_genes, 2023, doi:10.3390/genes14091732_

Round 1

Reviewer 1 Report

In this MS, the authors provided the experimental evidence that ZmOPR2, a member of OPRI subfamily enzymes, modulates defense against leaf herbivores by regulating JA homeostasis, and ZmOPR2 and ZmLOX10 synergistically contribute to insect defense in the leaves. In contrast, ZmOPR2 and ZmLOX10 cooperatively suppress resistance to WCR by inhibiting synthesis of death acids and ketols in roots upon WCR infestation. The results suggest the roles of these metabolites in defense against root herbivory. The experiment designs are feasible, the data and results are scientifically robust. The results are informative for the understanding the mechanism of plant defense against insect damage. I have only one minor comment as following: 

P6 line 231: Colletotrichum graminicola should be in italic

Author Response

We would like to thank Reviewers for taking the necessary time and effort to review the manuscript. Colletotrichum graminicola in Page 6 line 241 is italic.

Reviewer 2 Report

The fall armyworm (Spodoptera frugiperda) and western corn rootworm (Diabrotica virgifera virgifera) pose significant threats to maize crops. Essential in insect defense, jasmonic acid (JA) plays a pivotal role. The conversion of 12-oxo-phytodienoic acid (12-OPDA) to JA is facilitated by specific enzymes. In this investigation, the authors concentrated on the activities of distinct enzymes in this conversion process. They made the noteworthy discovery that ZmOPR2 and ZmLOX10 regulate JA-mediated resistance in leaves against FAW, while concurrently curbing the synthesis of insecticidal oxylipins in roots during WCR infestations. The research was methodically executed, yet a few questions listed:

The suppression of OPR2 expression compromised defense against FAW, whereas OPR2 overexpression bolstered defense. However, the dual mutants of lox10 and OPR2 displayed even greater susceptibility than single mutants. Does the overexpression of LOX10 similarly enhance defense against insects?

While the study primarily delves into the direct impacts of ZmOPR2 and ZmLOX10 on defense mechanisms, other potential interactions remain unexplored. Please include information regarding genetic alterations resulting from OPR2 overexpression and OPR2 or LOX10 knockout, extending beyond immediate defense mechanisms to broader plant physiology.

In the title, substituting ZmOPR2 and ZmLOX10 with "maize OPR2 and LOX10" might be more appropriate. Furthermore, the full name of WCR should be provided in the Abstract.

Notably, inconsistencies in figure color and resolution exist, particularly in Fig 5 and 6. Ensuring uniformity in color and pattern presentation is advised. Given the abundance of sample dots, reconsidering the necessity of error bars could be worthwhile.

Author Response

  1. The suppression of OPR2 expression compromised defense against FAW, whereas OPR2 overexpression bolstered defense. However, the dual mutants of lox10 and OPR2 displayed even greater susceptibility than single mutants. Does the overexpression of LOX10 similarly enhance defense against insects?

The role of maize LOX10 in defense against chewing fall armyworm and beet armyworm, has been well established in Christensen et al. (2013) and Yuan et al. (2023). Enhanced defense responses are expected while overexpressing maize LOX10 in Arabidopsis though the ZmLOX10 overexpression line is not available.

  1. While the study primarily delves into the direct impacts of ZmOPR2 and ZmLOX10 on defense mechanisms, other potential interactions remain unexplored. Please include information regarding genetic alterations resulting from OPR2 overexpression and OPR2 or LOX10 knockout, extending beyond immediate defense mechanisms to broader plant physiology.

Overexpression ZmOPR2 or disruption of ZmOPR2 and ZmLOX10 did not result in significant change in plant growth or development and no significant change in expression levels of stress-related genes at resting state in opr2 or lox10 mutants have been shown in Huang et al. (2023) and Yuan et al. (2023). Both ZmOPR2 and ZmLOX10 regulate stress-specific responses and gene expressions have also been addressed in these two papers.

  1. In the title, substituting ZmOPR2 and ZmLOX10 with "maize OPR2 and LOX10" might be more appropriate. Furthermore, the full name of WCR should be provided in the Abstract.

Thank you and we have made these changes according to your suggestions.

  1. Notably, inconsistencies in figure color and resolution exist, particularly in Fig 5 and 6. Ensuring uniformity in color and pattern presentation is advised. Given the abundance of sample dots, reconsidering the necessity of error bars could be worthwhile.

Thank you and we have ensured that the same colors are uniformly used in the figures. Also, the standard errors are included in all the figures.